# The Impact of the Degree of Urbanization on Spatial Distribution, Sources and Levels of Heavy Metals Pollution in Urban Soils—A Case Study of the City of Belgrade (Serbia)

**Mirjana Tešić [1],*** [iD], **Nadežda Stojanović [1]** [iD], **Milan Knežević [2]**, **Danijela Đunisijević-Bojović [1]**, **Jovana Petrović [1]** and **Pavle Pavlović [3]**

1   Department of Landscape Architecture and Horticulture, Faculty of Forestry, University of Belgrade, 11030 Belgrade, Serbia
2   Department of Forestry, Faculty of Forestry, University of Belgrade, 11030 Belgrade, Serbia
3   Department of Ecology, Institute for Biological Research 'Siniša Stanković'—National Institute of the Republic of Serbia, University of Belgrade, 11000 Belgrade, Serbia
*   Correspondence: mirjana.mesicek@sfb.bg.ac.rs

**Abstract:** This study investigated the effects of urbanization on the spatial distribution, sources, and levels of heavy metals pollution in urban soils of the city of Belgrade. A total of 126 composite topsoil (0–10 cm) and subsoil (10–40 cm) samples was collected within four urban zones (central, suburban, external, and rural) of Belgrade and analysed for content, spatial distribution, sources, and pollution indices using statistical methods including descriptive statistics, correlation matrix, and Principal Component Analysis (PCA). Concentration levels of zinc (Zn), copper (Cu), cadmium (Cd), chromium (Cr), and nickel (Ni) in the soil of urban zones in Belgrade were found to be significantly higher in the central and suburban zones compared to the external and rural ones. The spatial variance and the general trend of heavy metals accumulation in the soil were found to be along the urban–rural gradient. The topsoil concentration levels for the same elements were higher compared to the subsoil concentration levels for the same elements in all urban zones, except for the external and the rural zones. These results indicate the need for the implementation of urban soils pollution monitoring according to specific urban zones to provide an applicable basis for the development of plans and strategies concerning urban soil use management for the purpose of the sustainable urban development.

**Keywords:** urban soils; heavy metals; soil pollution; urban–rural gradient; sources of pollution; urban ecology

## 1. Introduction

One of the main consequences of intensive urbanization, which is nowadays a significant issue in many cities worldwide, is the continuous pollution of the urban environment with a myriad of pollutants that, in most cases, are not biodegradable, may be toxic and often accumulate in high concentrations, especially in urban soils and plants [1]. Past and current research show that pollutants, in significant proportions, primarily occur and accumulate in surface layers of the soil, thus affecting its characteristics [2]. Urban soils are exposed to continuous pollution with different contaminants, among which heavy metals are most frequently the dominant ones [3–5]. The pollution sources of heavy metals in the urban environment are mainly derived from anthropogenic sources. In urban soils and urban road dusts, the anthropogenic sources of heavy metals include traffic emissions (vehicle exhaust particles, tyre wear particles, weathered street surface particles, brake lining wear particles), industrial emissions (power plants, coal combustion, metallurgical industry, chemical plants, etc.), domestic emissions, weathering of building and pavement surfaces, atmospheric deposition, etc. [6–8]. Heavy metals have a negative impact on all

living organisms, and particularly on human health, as through soil and plants they can penetrate food chains and directly affect humans [9], therefore, studies on the presence and accumulation of heavy metals are also significant from the aspect of human health risk assessment [10,11]. The presence of heavy metals in urban ecosystems impacts the overall quality of the urban environment [12]. Therefore, the analysis of heavy metal concentrations in soils and their sources is required to identify levels of pollution and their sources. There are two main sources of heavy metals in urban soils: geogenic/lithogenic and anthropogenic [13]. One of the main differences between the two is in the level of pollution, where the levels of heavy metals pollution from the latter are higher than those from the former [14]. Furthermore, heavy metals from anthropogenic sources can be more mobile compared to their forms from geological ones [15,16]. In addition, heavy metals are not biodegradable, so they remain in the soil for a long period of time, and therefore, potentially pose a greater risk to human health and the urban environment [17]. However, despite the degradation of urban soil in general caused by soil pollution, fragmentation, mixing of soil layers, different substrates, etc., these soils can still support a wide range of urban functions [3,5,13,17–19]. These urban functions primarily refer to the soil as support for the green infrastructure that represents a coherent network of open green areas in a city as the most optimal model for the development of resilient cities [20]. This suggests that the approach to urban planning and management for the purpose of preservation of the urban environment must consider altered characteristics of urban soils and the degree of their pollution [12].

Metal concentrations in urban topsoils can be strongly influenced by land use, e.g., by residential, commercial, industrial, recreational, and even agricultural functions [21]. Therefore, intervention criteria to assess the pollution of the soil surface are defined according to the type of land use [22]. Previous research on the distribution of urban soil pollution has shown a significant relationship between the pollution intensity and the degree of urbanization (urban development), i.e., it was observed that the impact of urbanization in urban areas is primarily reflected in urban soils pollution along the urban–rural gradient [23,24]. The urban and rural environments are two central concepts that are widely used by urban planners and managers, researchers, (*national administrations*) and international organizations such as *the Organization for Economic Cooperation and Development* (OECD), the United Nations and the European Union [25]. An intensive transformation of an urbanized area results in different configurational types of urban development [26,27]. The formed patterns of this urban transformation represent an indicator and a powerful tool for the understanding of the development process of urban areas, the identification of the degree of urban expansion, and the prediction of further urban growth [28]. According to the intensity of urbanization, i.e., the degree of development of an urban area, many cities are divided into specific urban zones. Urban zones are operational planning units, separate parts of the city territory with similar characteristics regarding the presence of built-up areas. This uniformity enables them to be used as applicable tools for a plethora of research on the impact of the degree of urbanization on the urban environment. The concept of urban zones, in this essence, is closely connected with two basic factors: the urban landscape environment and human activity patterns [29]. The former provides the fundamental space for human activities and impacts the way urban soil is used, while the latter implies the differentiation of urban functions within these basic spatial units. In the areas of sustainable urban development and protection of the urban environment, the recognition of existing urban zones is more than helpful for understanding the spatial structure of a city and guiding the configuration of resources, which especially improves the governance mechanism for their future planning and management [30].

Changes consequent to urbanization play a significant role in the composition of urban soils in cities [31]. Cities and urban processes have had dramatic but varying impacts on the soil's physical and chemical properties and pollutant loads, all of which affect the life-supporting functions of soils. As developing countries, such as Serbia, continue to industrialize, soil pollution in their cities continues to increase to the levels that call for

immediate action. There is also the need for soil protection in the areas already undergoing changes because of urban development [32]. Soil pollution in the urban–rural zone has seriously affected the quality of urban environments and has restricted sustainable development [33]. To achieve science-based urban planning and sustainable development, it is necessary to understand the urban spatial structure and the functions of urban zones [29,34]. As the soils are long-term repositories of many elements that are delivered by atmospheric deposition, their characteristics represent an integration of many processes of degradation and pollution during the long history of urbanization and industrialization, as is the case with the city of Belgrade.

In this regard, this study aimed to: (1) investigate heavy metal accumulation in the topsoil and subsoil; (2) determine the distribution patterns of heavy metals pollution in urban soils; (3) analyse the degree of heavy metals pollution in urban soils in relation to the level of urbanization (pertaining to a specific urban zone) by means of pollution indices (Single Pollution Index (PI), Nemerov Pollution Index ($PI_{Nemerov}$), Pollution Load Index (PLI), Potential ecological risk (RI), for the purpose of the urban environment protection of the city of Belgrade (Serbia). The hypotheses were as follows: (1) the distribution of heavy metals and their pollution levels (indices) (in topsoil and subsoil) depend on the level of urbanization (pertaining to a specific urban zone), and it changes along the urban–rural gradient; (2) there are differences in the heavy metals distribution in urban soils according on soil depth in relation to the level of urbanization (pertaining to a specific urban zone); (3) the sources (natural and/or anthropogenic) of heavy metals pollution of the soils of the city of Belgrade are different depending on the city's urban zone. The results of this research provide recommendations and guidelines for appropriate planning and management of urban soil use as well as improvements of its general quality for the purpose of the urban environment protection in the city of Belgrade.

## 2. Materials and Methods

### 2.1. Study Area

The research was carried out in the territory of the city of Belgrade (44°49′ N and 20°27′ E). The surface area of Belgrade was 322,268 ha, while the specific urban area amounted to 35,996 ha. The population of the city of Belgrade was around 1,700,000 registered inhabitants in 2019. The expansion of urban spatial and functional systems of cities in the Republic of Serbia has been induced by changes in the economic activities and occupations of citizens in the settlements that are located near the urban centres. These settlements have eventually grown together and consequently have merged administratively. In this way, the urban fabric of the city of Belgrade has expanded by transforming the surrounding rural settlements where industrial zones have been developed, industrial facilities have been located, residential areas have been constructed and the capacity of the infrastructure and super-structure facilities has been increased. This has led to a great increase in the administrative territory of the city of Belgrade, which now occupies 3.6% of the total territory of the Republic of Serbia and where nearly a quarter of the total population of the country (23.1%) and almost a third of its total urban population lives [35]. A very dense transportation infrastructure is a major characteristic of the city of Belgrade, making transportation (along with economy and industry) one of the main sources of pollution of the urban environment. Occupying significant surfaces of urban soils for housing purposes has caused a radial concentric development of the city in comparison to the traditional city centre concept. Such spatial expansion of the city has brought about a disbalance in the urban soil use planning and in the redistribution of urban soils functions.

The Master Plan for the city of Belgrade 2021 [36] defines four urban zones (units) that make up the main spatial-functional, urban-architectural, and landscape-ecological zones of the city of Belgrade, namely: the central, the suburban, the external, and the rural zones. The central zone of the city of Belgrade consists of the city's urban core (city centre) with different types of man-made (artificial) elements. Administrative, residential, and cultural areas (with a high population density) are included here, with a dense built

infrastructure, including buildings, roads, pavements, bridges, city squares and parks, etc. The suburban zone represents a wider area of the built-up part of the city, which includes built-up and commercial city areas, large areas of impervious surfaces, recreational and sports facilities, roads, industrial areas, urban parks, etc. The external zone consists of agricultural, abandoned, and semi-natural areas, with a low population density. The rural zone consists of natural and protected areas and other natural systems (urban forests, groves, greenery of watercourses, urban wetlands, etc.) [36].

The geological substrate of the city of Belgrade comprises sedimentary rocks (loess, clay, slate, sandstone, marl, limestone, sand, etc.), metamorphic and magmatic rocks (primarily serpentine) [37], which has caused the occurrence and development of different types of soils in this area: colluvium, calcomelanosol, regosol, lithosol, chernozem, eutric cambisol, fluvisol, humogley and eugley [38]. In this area the natural vegetation includes climatogenic forests of Italian and Austrian Oak (*Quercetum frainetto—Q. cerris* Rud. 1940), azonal vegetation of alluvial forests of white willow (*Salicion albae* Soo 1940), hygrophilous forests of Alder and European Oak (*Alno—Quercion roboris* Horv. 1938) associations [39] and forest-steppe communities (*Andropogoneto — Euphorbietum panonnicae* R. Bog. and *Querceto—Carpinetum serbicum* Rudski 1940) [40]. The city of Belgrade is located at the altitude of 116 m and has a moderate continental climate. Its average annual air temperature is 12.5 °C (the temperature ranges from –1.1 °C in January to 23 °C in July). According to the climatic data, there are, on average, 31 days a year when the temperature is above 30 °C (the so-called tropical days) and 95 summer days, when the temperature is above 25 °C. The average annual amount of precipitation is 690 mm/year [41]. In Belgrade, the dominant wind is from the south and southeast, and from the west and northwest is also frequent.

### 2.2. Soil Sampling and Laboratory Analysis

The topsoil (0–10 cm depth) and subsoil (10–40 cm depth) samples were randomly collected in the four urban zones of the city of Belgrade as follows: 52 in the central zone, 46 in the suburban zone, 16 in the external zone, and 12 samples in the rural zone. A total of 126 composite disturbed soil samples were collected from the topsoil depth of 0–10 cm and the subsoil depth of 10–40 cm (Figure 1).

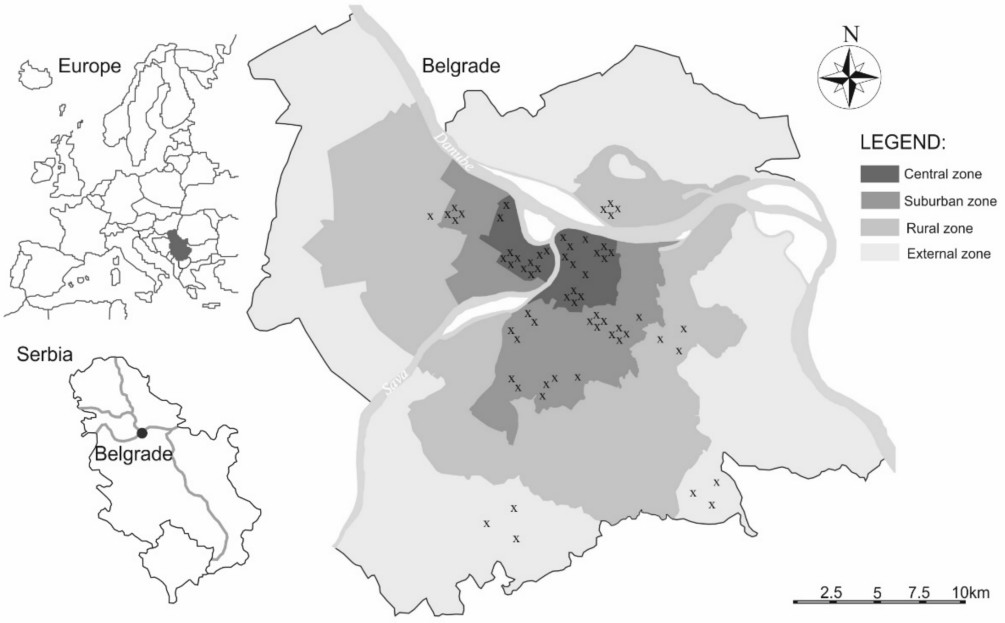

**Figure 1.** Belgrade urban zones and soil sampling sites/locations.

The composite samples were taken following the procedure outlined by [4]. The soil was sampled by means of a stainless soil auger. One composite sample was taken in the locations whose surface area exceeded 4 ha and where the biggest percentage of the soil (>50%) was under inert, anthropogenic materials (pathways, roads, buildings, etc.). Three composite soil samples were collected in the locations with self-grown vegetation (the locations in the external zone and the rural zone).

The laboratory analysis of the soil was carried out in the Laboratory for Soil Quality Monitoring of the Faculty of Forestry, University of Belgrade. Stones, roots, and other recognizable parts of plants were removed from the samples and the soil was dried and screened through 2 mm sieves. The total concentration levels of heavy metals Zinc (Zn), Copper (Cu), Cadmium (Cd), Lead (Pb), Chromium (Cr) and Nickel (Ni) in the soil were determined by means of the AAS (Atomic Absorption Spectroscopy) method using a *Thermo M Series A*, USA, instrument, while the preparation of samples was performed by Royal Water digestion [42]. The soil samples were digested with aqua regia under reflux for 2 h, with water-cooled condensers, to determine the content of trace elements [43]. The content of heavy metals (Zn, Cu, Cd, Pb, Cr, and Ni) was determined by using flame atomic absorption spectrophotometry (AAS) [44]. The chemical analyses were performed in two replications. For the verification of results, the referent soil was examined for the presence of microelements (reference ERM-CC141 loam soil, Belgium, with the exact concentration of microelements soluble in aqua regia to provide for increased accuracy of the measuring apparatus). The lower limits of detection for the elements were as follows (mg/kg): Zn—0.01679, Cu—0.06549, Cd—0.01798, Pb—0.11111, Cr—0.03203, and Ni—0.034302.

For evaluation of the impact of urbanization on the distribution and the degree of heavy metals pollution of soils in the different urban zones of Belgrade, the measured mean values of concentrations of heavy metals were compared with the heavy metals concentration values of natural soils determined for the soils of Central Serbia (the reference value) [45,46]. In addition, the measured mean values for heavy metals concentrations in the soils of the urban zones of Belgrade were compared to their maximum limit values (MLV) and the remediation values, which when exceeded, point to the degree of pollution (contamination) that disrupts the environmental balance, according to *the Decree on Limit Values of Pollutants, Harmful and Hazardous Substances in Soil* used as a set of indicators for risk assessment related to soil degradation and as the methodology for the development of remediation programs in the Republic of Serbia [47] (Table 1).

**Table 1.** MLV, remediation and reference values of heavy metals (Zn, Cu, Cd, Pb, Cr and Ni) in soils of the Republic of Serbia [45–47].

| Heavy Metal | Concentration of Heavy Metals in Soils (mg/kg) | | | | | |
|---|---|---|---|---|---|---|
| | *Zn* | *Cu* | *Cd* | *Pb* | *Cr* | *Ni* |
| Background value | 68.14 | 31.8 | 0.86 | 54.92 | 52.40 | 52 |
| Maximum limit value | 140 | 36 | 0.8 | 85 | 100 | 35 |
| Remediation value | 720 | 190 | 12 | 530 | 380 | 210 |

*2.3. Soil Pollution Indices*

To analyse the impact of the degree of urbanization on the degree of soil pollution by heavy metals in the city of Belgrade, pollution indices were calculated for both soil layers according to [10,48–50]:

$$\text{Single Pollution Index} - \text{PI} = \frac{C_n}{GB} \tag{1}$$

where $C_n$ is the measured mean concentration of heavy metals; *GB* is the background value. It was used to determine which heavy metal posed the greatest danger to the soil environment. It is also necessary for the calculation of some of the complex indices: $\text{PI} \leq 1$ low; $1 \leq \text{PI} \leq 2$ moderate; $2 \leq \text{PI} \leq 5$ high; $\text{PI} \leq 5$ extremely high level of pollution.

$$\text{Nemerov Pollution Index} - \text{PI}_{\text{Nemerov}} = \sqrt{\frac{\left(\frac{1}{n}\sum_{i-1}^{n} PI\right)^2 + PI_{max}{}^2}{n}} \tag{2}$$

where *PI* is the calculated value of the single pollution index, *PI_{max}* is the maximum value of PI of all analysed heavy metals: *n* is the number of analysed heavy metals. It enables the assessment of the overall degree of soil pollution and includes the concentration of all analysed heavy metals: the I class ≤0.7, clean; II class 0.7–1, warning limit; III class 1–2, slight pollution; IV class 2–3, moderate pollution; V ≥ 3 heavy pollution.

$$\text{Pollution Load Index} - \text{PLI} = \sqrt[n]{PI_1 \; x \; PI_2 \; x \; \dots \; PI_n} \tag{3}$$

where *n* is the number of analysed heavy metals; *PI* is calculated value of the single pollution index. It was used for the overall assessment of the degree of soil pollution: <1 unpolluted land; 1 basic level of pollution; >1 deterioration of the soil quality.

$$\text{Single index of the ecological risk} - E_r^i = T_r^i \times \text{PI} \tag{4}$$

where *Tr* is the toxicity response coefficient of an individual metal; *PI* is the value of the single pollution index. It was used to calculate RI. According to Håkanson [48], the *Tr* for heavy metals is: Zn = 1, Cr = 2, Cu, Pb and Ni = 5 and Cd = 30.

$$\text{Potential ecological risk} - \text{RI} = \sum_{i=1}^{n} E_r^i \tag{5}$$

where *n* is the number of heavy metals; *Er* is the single index of the ecological risk. The RI is an index applicable to assess the degree of environmental risk caused by elevated concentrations of heavy metals in water, air, and soil: <90 low; 90–180 moderate; 180–360 strong; 369–720 very strong and >720 very strong potential environmental risk.

### 2.4. Statistical Data Analysis

The processing of statistical data was carried out using the software suite SPSS 20. The differences in mean values of concentrations of the analysed heavy metals (Zn, Cu, Cd, Pb, Cr and Ni) in topsoil (0–10 cm) and subsoil (10–40 cm) of the four urban zones of Belgrade (central, suburban, external, and rural urban zones) were determined by means of descriptive statistical analysis. The correlation matrix and the Principal Component Analysis (PCA) were used to determine the sources of the analysed heavy metals by reducing the dataset dimensionality to several significant factors [51–53]. Varimax factor rotation with Kaiser normalization was used for carrying out the PCA. The ArcGIS version 10.8.2. (Belgrade, Serbia) software was used for the graphic presentation of the distribution of heavy metals in the soils of urban zones of the city of Belgrade as well as for the graphic presentation of pollution indices.

### 3. Results

### 3.1. The Distribution of Zn, Cu, Cd, Pb, Cr and Ni in the Soils of Urban Zones of the City of Belgrade

The obtained mean values of concentrations of heavy metals (Zn, Cu, Cd, Pb, Cr and Ni) in the soils of four urban zones of the city of Belgrade are shown in Table 2.

#### 3.1.1. Zinc Concentrations

The highest mean concentration values of Zn were measured in the topsoils of the suburban zone (154.67 ± 119.06 mg/kg) and the central zone (141.15 mg/kg). Both measured mean values exceeded the MLV (140 mg/kg) and reference (68.14 mg/kg) values. The mean concentration values of Zn decreased with the depth of the soil. The mean concentration values of Zn in the subsoil followed the same pattern of the mean concentration values of Zn in the topsoil. The highest mean concentration values of Zn in these soil layers were measured in the suburban (116.75 ± 85.04 mg/kg) and the central (99.41 ± 33.02 mg/kg) zones. The mean concentration values of Zn in the subsoils of the external and rural zones

were lower than the MLV, and slightly higher than the reference value. These results concurred with earlier findings of Lee et al. [54] on the higher concentration of Zn in the soils of the urban zone of Hong Kong (103 mg/kg) compared with its suburban (67.9 mg/kg) and rural (46.8 mg/kg) zones. Additionally, the research by Rodrigues et al. [55] focusing on the soils of the city of Porto (Portugal), and the research by Zou et al. [56] focusing on the soils of the city of Shijiazhuang (China), confirmed that the increase in the concentration of Zn moves along the urban–rural gradient, i.e., its highest concentrations were recorded in the soils of the urban zone of the city when compared to those in the rural zones.

**Table 2.** Mean (±SD) topsoil and subsoil heavy metals (Zn, Cu, Cd, Pb, Cr and Ni) concentration values in different urban zones of the city of Belgrade (mg/kg).

| HM. | Urban Zone | $n$ | Topsoil (0–10 cm) | | | | Depth (10–40 cm) | | | |
|---|---|---|---|---|---|---|---|---|---|---|
| | | | Mean | SD | Min. | Max. | Mean | SD | Min. | Max. |
| Zn | Central zone | 26 | 141.15 | 67.72 | 54.95 | 339.86 | 99.41 | 33.02 | 47.26 | 176.06 |
| | Suburban zone | 23 | 154.67 | 119.06 | 63.30 | 469.12 | 116.75 | 85.04 | 60.69 | 434.86 |
| | External zone | 8 | 76.59 | 30.30 | 35.21 | 141.23 | 71.88 | 27.98 | 49.31 | 138.69 |
| | Rural zone | 6 | 68.59 | 11.57 | 59.83 | 90.47 | 62.13 | 3.12 | 59.86 | 68.21 |
| | Total | 63 | 130.98 | 89.46 | 35.21 | 469.12 | 98.69 | 58.64 | 47.26 | 434.86 |
| Cu | Central zone | 26 | 65.88 | 75.34 | 13.91 | 422.42 | 43.10 | 16.25 | 16.37 | 78.56 |
| | Suburban zone | 23 | 42.92 | 27.55 | 15.87 | 152.84 | 35.53 | 12.99 | 18.32 | 64.02 |
| | External zone | 8 | 21.64 | 7.89 | 13.36 | 39.33 | 20.98 | 8.91 | 6.97 | 38.39 |
| | Rural zone | 6 | 19.06 | 4.52 | 15.91 | 28.04 | 19.15 | 2.43 | 15.15 | 22.62 |
| | Total | 63 | 47.42 | 53.67 | 13.36 | 422.42 | 35.25 | 15.91 | 6.97 | 78.56 |
| Cd | Central zone | 26 | 0.86 | 0.76 | 0.00 | 2.77 | 0.64 | 0.38 | 0.00 | 1.31 |
| | Suburban zone | 23 | 0.85 | 0.76 | 0.00 | 2.47 | 0.61 | 0.66 | 0.00 | 2.59 |
| | External zone | 8 | 0.35 | 0.41 | 0.00 | 1.16 | 0.35 | 0.35 | 0.00 | 0.94 |
| | Rural zone | 6 | 0.39 | 0.19 | 0.19 | 0.58 | 0.42 | 0.19 | 0.24 | 0.68 |
| | Total | 63 | 0.75 | 0.71 | 0.00 | 2.77 | 0.58 | 0.49 | 0.00 | 2.59 |
| Pb | Central zone | 26 | 167.35 | 257.15 | 14.03 | 1308.38 | 102.76 | 169.40 | 15.96 | 898.13 |
| | Suburban zone | 23 | 82.33 | 103.68 | 20.97 | 472.35 | 56.33 | 62.61 | 15.86 | 299.71 |
| | External zone | 8 | 38.52 | 37.01 | 12.99 | 124.38 | 24.51 | 12.63 | 10.12 | 50.42 |
| | Rural zone | 6 | 32.71 | 11.19 | 15.50 | 42.73 | 36.72 | 6.35 | 31.45 | 48.81 |
| | Total | 63 | 107.13 | 183.12 | 12.99 | 1308.38 | 69.58 | 117.80 | 10.12 | 898.13 |
| Cr | Central zone | 26 | 62.24 | 24.74 | 16.66 | 121.58 | 60.56 | 19.52 | 28.06 | 99.00 |
| | Suburban zone | 23 | 62.84 | 18.48 | 28.93 | 107.37 | 58.10 | 18.90 | 32.76 | 107.65 |
| | External zone | 8 | 49.62 | 16.74 | 22.29 | 68.15 | 41.92 | 20.34 | 13.88 | 73.77 |
| | Rural zone | 6 | 50.08 | 18.98 | 22.21 | 70.90 | 44.43 | 6.88 | 37.48 | 53.08 |
| | Total | 63 | 59.70 | 21.38 | 16.66 | 121.58 | 55.76 | 19.49 | 13.88 | 107.65 |
| Ni | Central zone | 26 | 56.25 | 14.64 | 31.57 | 96.75 | 59.47 | 19.02 | 32.57 | 108.88 |
| | Suburban zone | 23 | 57.22 | 18.74 | 33.43 | 116.16 | 60.01 | 23.12 | 39.03 | 141.36 |
| | External zone | 8 | 48.92 | 19.04 | 22.62 | 82.76 | 49.13 | 25.31 | 15.77 | 99.24 |
| | Rural zone | 6 | 37.15 | 12.41 | 23.25 | 52.75 | 38.48 | 9.47 | 32.19 | 57.17 |
| | Total | 63 | 53.85 | 17.35 | 22.62 | 116.16 | 56.36 | 21.49 | 15.77 | 141.36 |

### 3.1.2. Copper Concentrations

The mean concentration values of Cu in the topsoils increased from the rural (19.06 ± 4.52 mg/kg) towards the central zone of the city (65.88 ± 75.34 mg/kg). The mean concentration values of Cu in all analysed soils were within the allowed limits, in line with the Decree on Limit Values of Pollutants, Harmful and Hazardous Substances in Soil [47], while the ones in the topsoils of the central (65.88 ± 75.34 mg/kg) and the suburban (42.92 ± 27.55 mg/kg) zones of the city were higher than the reference values (31.80 mg/kg). The mean concentration values of Cu in the subsoils followed the pattern obtained in the topsoils, thus ranging from the lowest in the rural zone (19.15 ± 2.43 mg/kg) to the highest in the central zone of the city (43.10 ± 16.25 mg/kg). As in the topsoils, the mean concentration values of Cu in the subsoils of the central (43.10 ± 16.25 mg/kg)

and the suburban ($35.53 \pm 19.99$ mg/kg) zones of the city were higher than the reference value. These data were also in accordance with the earlier research findings by Lee et al. [54] focusing on the soils of Hong Kong (China), that showed that the mean concentration values of Cu were the highest in the soils of the urban zone (16.2 mg/kg), lower in the suburban zone (9.72 mg/kg) and the lowest in the rural zone (6.37 mg/kg) of the city. Moreover, the research by Mao et al. [57] focusing on the soils of the city of Beijing (China) showed that the mean concentration values of Cu were the highest in the soils of the urban zone (39.95 mg/kg) and that they decreased with distance from the centre of the city towards the transitional (28.98 mg/kg) and suburban (25.78 mg/kg) zones. The research focusing on the soils of different urban zones of the city of Torino (Italy) also showed that the mean concentration value of Cu in the urban soils (90 mg/kg) was higher than in the rural ones (28 mg/kg) [23].

### 3.1.3. Cadmium Concentrations

The expected highest mean concentration values of Cd were measured in the topsoils of the central zone ($0.86 \pm 0.76$ mg/kg) and the suburban zone ($0.85 \pm 0.76$ mg/kg) of the city. However, the lowest mean concentration value of Cd was measured in the topsoils of the external zone ($0.35 \pm 0.41$ mg/kg) and a slightly higher mean value was measured in the rural zone ($0.39 \pm 0.19$ mg/kg) of the city. In the subsoils of the urban zones of the city of Belgrade, the measured mean concentration values of Cd were in the same descending order as those measured in the topsoils: in the central zone ($0.64 \pm 0.38$ mg/kg), in the suburban zone ($0.61 \pm 0.66$ mg/kg), in the rural zone ($0.42 \pm 0.19$ mg/kg), and the external zone ($0.35 \pm 0.41$ mg / kg) of the city. Only the mean concentration value of Cd in the topsoil of the central zone exceeded the MLV (0.8 mg/kg) and was at the maximum limit of its reference value (0.86 mg/kg). It should be noted that the mean concentration value of Cd in the topsoil of the suburban zone was also at the maximum limit of its reference values. A slightly higher mean concentration value of Cd measured in the subsoil of the rural zone of the city was associated with features of the geological substrate, which consists mainly of sedimentary, and in some places of metamorphic and magmatic rocks [58]. These results are in accordance with the results of the research into the mean concentration values of Cd in the soils of the city of Porto (Portugal), which showed the highest concentrations in the urban zone (0.43 mg/kg) and significantly lower ones in the rural zone (0.20 mg/kg) [55]. These results are also partly in accordance with the results of the research into the urban soils of the city of Vienna, which showed that the mean concentration values of Cd were almost identical in the urban zone (0.2 mg/kg) and the suburban zone (0.2 mg/kg) and higher than those of the rural zone (0.1 mg/kg) [49]. As in this research, the mean concentration values of Cd in the soils of different urban zones of the city of Belgrade incline towards the urban–rural gradient, but here the mean concentration values of Cd were significantly higher: in the central zone (0.86 mg/kg), in the suburban zone (0.78 mg/kg), in the rural zone (0.39 mg/kg) and in the external zone (0.33 mg/kg). It is also important to note that Kabata-Pendias [59] stated that Cd > 0.5 mg/kg concentrations are of anthropogenic origin, which indicates that the measured mean concentration values of Cd in the soils of the external zone and the rural zone of the city of Belgrade are of geological origin as they do not exceed this value. However, the maximum concentration values of Cd measured in the soils of these zones in both layers were more than >0.5 mg/kg, which leads to the conclusion that the origin of Cd in these soils is partly geological and partly anthropogenic, originating from atmospheric deposition.

### 3.1.4. Lead Concentrations

The mean concentration values of Pb in the topsoils increased from the rural ($32.71 \pm 11.19$ mg/kg) towards the central ($167.35 \pm 257.15$ mg/kg) zone of the city where they exceeded MLV (85 mg/kg) and the reference value (54.92 mg/kg). The mean concentration value of Pb in the topsoils of the suburban zone of the city ($82.33 \pm 103.68$ mg/kg) also exceeded the reference value. In the subsoils and in the topsoils, the highest mean

concentration values of Pb were measured in the central zone (102.76 ± 169.40 mg/kg) and in the suburban zone (56.33 ± 62.61 mg/kg) of the city, however, unlike the topsoils, it declined in the soils of the rural zone (36.72 mg/kg), while the lowest mean concentration value was measured in the subsoils of the external zone of the city (24.51 ± 12.63 mg/kg). As for the topsoils, only the mean concentration values of Pb in the subsoil of the central zone exceeded its MLV (85 mg/kg) and the reference value (54.92 mg/kg). The mean concentration value of Pb in the subsoils of the city's suburban zone was at the upper limit of its reference value (54.92 mg/kg). These results are in accordance with the findings of Rodrigues et al. [55] that established significantly higher concentrations of Pb in the soils of the urban zone (137 mg/kg, rang 27–430 mg/kg) in the city of Porto (Portugal) compared to those in the rural zone (35 mg/kg, rang 11–86 mg/kg), and are also in accordance with the findings of Foti et al. [24], where the soils of the city of Paris (France) showed a significantly higher concentration of Pb in the urban zone (130.19 mg/kg) compared to those in the suburban (75.98 mg/kg) and the rural (36.05 mg/kg) zones. According to the findings of Lee et al. [54], the city of Hong Kong also showed a higher concentration of Pb in the urban soils (88.1 mg/kg) compared to the soils in the suburban (57.8 mg/kg) and the rural (39.6 mg/kg) zones.

### 3.1.5. Chromium Concentrations

The highest mean concentration value of Cr was measured in the topsoils of the suburban zone (62.84 ± 18.48 mg/kg), with slightly lower concentrations in the topsoils of the central zone (62.24 ± 24.74 mg/kg). The mean concentration values of Cr declined with soil depth. The highest mean concentration value of Cr was measured in the subsoil of the central zone of the city (60.56 ± 19.52 mg/kg); a somewhat lower concentration was measured in the subsoils of the suburban zone (58.10 ± 18.90 mg/kg), followed by the rural zone (44.43 ± 6.88 mg/kg), and the lowest concentration was found in the subsoils of the external zone of the city (41.92 ± 20.34 mg/kg). The mean concentration values of Cr in both soil layers of all the analysed urban zones of the city were not higher than MLV, while the mean concentration values of Cr in the soils of the central and the suburban zones in both soil layers were higher than its reference value (52.40 mg/kg). Biasioli et al. [23] published similar results, stating that the soils of the urban zone of the city of Turin (191 mg/kg) had a significantly higher concentration of Cr than those in the rural zone (90 mg/kg). These data are also in accordance with the findings of the research focusing on other cities such as Paris (France), where the highest mean concentration of Cr was in the soils of the suburban zone (19.91 mg/kg), followed by that the urban zone (16.96 mg/kg), with the lowest in the soils of the rural zone (14.42 mg/kg) [24].

### 3.1.6. Nickle Concentrations

The mean concentration values of Ni in the topsoils of different urban zones of the city of Belgrade were determined and sorted in descending order: suburban (57.22 ± 18.74 mg/kg), central (56.25 ± 14.64 mg/kg), external (48.92 ± 19.04 mg/kg), and rural (37.15 ± 12.41 mg/kg) zones. The mean concentration values of Ni in the subsoils followed the pattern of the concentration values in the topsoils, thus the determined values were sorted in the same descending order: suburban (60.01 ± 23.12 mg/kg), central (59.47 ± 19.02 mg/kg), external (49.13 ± 25.31 mg/kg), and rural (38.48 ± 9.47 mg/kg). It was found that the mean concentration values of Ni increased with soil depth in all soils of the urban zones of the city of Belgrade. This result was associated with the features of the geological substrate (especially that of the rural zone), which is diverse, containing predominantly sedimentary rocks. The increased mean concentration values of Ni determined in the soils of the central zone, the suburban zone and the external zone are the result of anthropogenic pollution in all urban zones. This claim is confirmed by the research focusing on the soils and the sediments of the Sava River basin in the Belgrade area [53], which has showed that the increased concentrations of Ni (in addition to their geological sources) are also due to the anthropogenic pollution in this area. These results are partly in accordance with the results

of the research by Zou et al. [56] focusing on the soils of the city of Shijiazhuang (China), which showed that the highest concentration of Ni was in the soils of the northern part of the rural zone (28.2 mg/kg), followed by the urban zones (27.8 mg/kg), the suburban zone (21.4 mg/kg), and that the lowest concentration was in the soils of the southern part of the rural zone (20.5 mg/kg). The higher mean concertation values of Ni were also measured in the soils of the urban zone (209 mg/kg) of the city of Torino (Italy) and were significantly lower in the rural zone (74 mg/kg) of the city [23].

The results confirm that the mean concentration values of heavy metals in the soils of the city of Belgrade decreased with soil depth, except when it came to the concentrations of Cu, Cd and Pb in the soils of the rural zone of the city, while the mean concentration values of Ni increased with soil depth in all urban zones of the city of Belgrade, which was associated with the features of the geological substrate of the city area. Namely, in the central parts of Serbia, including Belgrade, the source of Ni and Cr in the soil was determined by the geological substrate. There are increased concentrations of Ni and Cr in the soil formed on serpentine rocks in the western Serbia and in the valleys of large rivers, where they have originated from paedogenetic processes of alluvium formation [46].

*3.2. Possible Sources of Heavy Metals (Zn, Cu, Cd, Pb, Cr and Ni) in the Soils of the Urban Zones of the City of Belgrade*

The possible sources of heavy metals (Zn, Cu, Cd, Pb, Cr and Ni) in the soils of the four urban zones of the city of Belgrade were determined based on the results of the correlation matrix and the PCA test.

The Pearson correlation coefficient was used to establish the relationship between heavy metals which are presented in Table 3.

**Table 3.** The Correlation matrix for heavy metals (Zn, Cu, Cd, Pb, Cr and Ni) in the soils of the urban zones of the city of Belgrade.

| HM | Zn | Cu | Cd | Pb | Cr | Ni | | Zn | Cu | Cd | Pb | Cr | Ni |
|---|---|---|---|---|---|---|---|---|---|---|---|---|---|
| Zn | 1 | | | | | | | 1 | 0.700 ** | 0.890 ** | 0.699 ** | 0.497 ** | 0.110 |
| Cu | 0.487 ** | 1 | | | | | | | 1 | 0.688 ** | 0.887 ** | 0.467 ** | 0.273 |
| Cd | 0.824 ** | 0.259 | 1 | | | | Suburban zone | | | 1 | 0.627 ** | 0.605 ** | 0.294 * |
| Pb | 0.293 * | 0.105 | 0.237 | 1 | | | | | | | 1 | 0.463 ** | 0.057 |
| Cr | 0.629 ** | 0.369 ** | 0.606 ** | 0.121 | 1 | | | | | | | 1 | 0.546 ** |
| Ni | 0.391 ** | 0.446 ** | 0.318 * | 0.074 | 0.713 ** | 1 | | | | | | | 1 |
| Zn | 1 | | | | | | | 1 | 0.698 * | 0.339 | −0.367 | 0.303 | 0.237 |
| Cu | 0.922 ** | 1 | | | | | | | 1 | 0.550 | −0.093 | 0.564 | 0.555 |
| Cd | 0.469 | 0.402 | 1 | | | | Rural zone | | | 1 | −0.174 | 0.120 | −0.050 |
| Pb | 0.828 ** | 0.812 ** | 0.450 | 1 | | | | | | | 1 | 0.176 | 0.082 |
| Cr | 0.513 * | 0.620 * | 0.198 | 0.490 | 1 | | | | | | | 1 | 0.807 ** |
| Ni | 0.192 | 0.358 | −0.212 | 0.218 | 0.852 ** | 1 | | | | | | | 1 |

(Left block zone labels: Central zone for first six rows, External zone for last six rows.)

** Correlation is significant at the 0.01 level (2-tailed); * Correlation is significant at the 0.05 level (2-tailed).

In soils of the central zone, Zn is correlated with Cu (r = 0.487 **), Cd (r = 0.824 **), Cr (r = 0.629 **), and Ni (r = 0.391 **); Cu with Cr (r = 0.369 **) and Ni (r = 0.446 **); Cd with Cr (r = 0.606 **), and Ni with Cr (r = 0.713 **). In soils of suburban zone, all heavy metals correlate with each other, except Ni which was correlated only with Cd (r = 0.294 **) and Cr (r = 0.546 **). In soils of the external zone, the Zn was correlated with Cu (r = 0.922 **) and Pb (r = 0.828 **), then Cu with Pb (r = 0.812 **) and Cr with Ni (r = 0.852 **). In soils of rural zone, the only correlation was between Cr and Ni (r = 0.807 **). These pairs and strong correlations may suggest a common origin at 99% confidence level.

The PCA results show that two main components combined described over 70% of the total variation, in four sets. The PCA in the central zone of the city of Belgrade showed that the first two components together described 69.82% of variability. The elements Cr and Ni were highly positively correlated in the first component (PC1) that explains 42.79% of

variability and Zn and Cu heavy metals were also significantly positively correlated. The second component (PC2) described 27.04% of variability and within this component Pb was highly positively correlated and Zn and Cd were significantly positively correlated (Table 4).

**Table 4.** Matrix of the principal component analysis loadings of heavy metals (Zn, Cu, Cd, Pb, Cr and Ni) in different urban zones of the city of Belgrade with cumulative percentage.

| Heavy Metals | Rotated Component Matrix [a] | | | | | | | |
|---|---|---|---|---|---|---|---|---|
| | Central Zone (*n* = 52) | | Suburban Zone (*n* = 54) | | External Zone (*n* = 8) | | Rural Zone (*n* = 12) | |
| | PC1 | PC2 | PC1 | PC2 | PC1 | PC2 | PC1 | PC2 |
| Zn | **0.63** | **0.66** | **0.90** | 0.13 | **0.91** | 0.25 | 0.31 | **0.79** |
| Cu | **0.67** | 0.08 | **0.877** | 0.20 | **0.85** | 0.41 | **0.66** | **0.66** |
| Cd | 0.53 | **0.68** | **0.83** | 0.34 | **0.75** | −0.28 | 0.06 | **0.72** |
| Pb | −0.12 | **0.81** | **0.91** | 0.03 | **0.866** | 0.25 | 0.31 | **−0.65** |
| Cr | **0.84** | 0.26 | 0.47 | **0.74** | 0.383 | **0.87** | **0.93** | 0.05 |
| Ni | **0.85** | −0.04 | −0.00 | **0.943** | −0.023 | **0.98** | **0.91** | −0.01 |
| Eigen-value | 2.57 | 1.62 | 3.32 | 1.61 | 3.00 | 2.09 | 2.33 | 2.00 |
| % variance explained | 42.79 | 27.04 | 55.38 | 26.91 | 49.997 | 34.85 | 38.85 | 33.25 |
| Cumulative % variance | 42.79 | 69.82 | 55.38 | 82.29 | 49.997 | 84.85 | 38.85 | 72.10 |

Extraction Method: Principal Component Analysis. Rotation Method: Varimax with Kaiser Normalization. [a] Rotation converged in three iterations.

In the suburban zone, 82.29% of variance described two components, where the first one was described by 55.38% and within it, Zn, Cu, Cd and Pb were highly correlated, while the other one was described by 26.91% and within it, Cr and Ni were highly correlated. In the external zone of the city, the results showed that the first two components described 84.85% of variability, where, as in the suburban zone, within the first component (PC1 49.99%), the heavy metals Zn, Cu, Cd and Pb were highly positively correlated, and in the second one (PC2), Cr and Ni (34.85%) were highly positively correlated. In the rural zone of the city, the first two components explained 72.10% and PC1 described 38.85% of variability, and within it, Cr and Ni were highly positively correlated with a significant correlation of Cu, while PC2 described 33.25% of the variability with significant correlation of Zn, Cu and Cd, and a negative correlation with Pb. Regarding these data, the PCA test results showed that Cr and Ni were presumably of the same geological source, and that Zn, Cu, Cd and Pb were of an anthropogenic source. However, regarding Cu, the PCA test showed that possibly it may be of a combined natural and anthropogenic source, because it was positively correlated in both components in the rural zone (Table 4).

### 3.3. Pollution Indices of the Soil in the City of Belgrade Relative to Its Urban Zones

Table 5 shows PI values of the soil of the urban zones in the city of Belgrade, as well as $PI_{Nemerov}$, PLI, and RI. According to the PI values, the soils (both soil layers) of the central zone belong to the class of low to moderately polluted soils, sorted in descending order: Pb (2.46) > Zn (1.77) > Cu (1.71) > Cr (1.17) > Ni (1.11) > Cd (0.87). In the suburban zone of the city, the PI values of all analysed heavy metals, except Cd, which belonged to the class of unpolluted soils, were in the soil class with low pollution indices, sorted in descending order: Zn (1.99) > Pb (1.26) > Cu (1.23) > Cr (1.15) > Ni (1.13) > Cd (0.85). In the soils of the external zone, the highest PI values were measured for Zn (1.09), which was the lower limit for the soils with low pollution, and for Ni (0.94) and Cr (0.87), which belonged to the class of unpolluted soils, although these values were at the upper limit of this class, while PI values for Pb (0.57), Cu (0.67) and Cd (0.41) classify the soils in this urban zone as unpolluted with these heavy metals. The PI values of all analysed heavy metals in the soils of the rural zone of the city showed that these soils were not polluted. It should be noted that the PI values for Zn (0.96) and Cr (0.90) were at the upper limit of this class.

**Table 5.** Indices of soil pollution of the analysed soils in the urban zones of the city of Belgrade.

| Urban Zone | PI Value | | | | | | PLI | PI$_{Nemerov}$ | RI |
|---|---|---|---|---|---|---|---|---|---|
| | **Zn** | **Cu** | **Cd** | **Pb** | **Cr** | **Ni** | | | |
| Central zone | 1.77 | 1.71 | 0.87 | 2.46 | 1.17 | 1.11 | 1.44 | 1.18 | 56.78 |
| Suburban zone | 1.99 | 1.23 | 0.85 | 1.26 | 1.15 | 1.13 | 1.40 | 0.73 | 47.84 |
| External zone | 1.09 | 0.67 | 0.41 | 0.57 | 0.87 | 0.94 | 1.29 | 0.39 | 26.70 |
| Rural zone | 0.96 | 0.60 | 0.47 | 0.63 | 0.90 | 0.73 | 1.27 | 0.49 | 28.40 |

The PI$_{Nemerov}$ values in the soils of the analysed urban zones of the city of Belgrade decreased, starting from the central zone, and moving towards the rural zone. According to the PI$_{Nemerov,}$ the soils of the central zone of the city belonged to the class of slightly polluted (PI$_{Nemerov}$ 1–2), the soils of the suburban zone belonged to the warning limit class, while the soils of the external and rural zones of the city belonged to the class of clean soils (PI$_{Nemerov} \leq 0.7$). The Pollution Load Indices (PLIs) also decreased, starting from the central zone, and moving towards the rural zone, however, in all urban zones they fell into the category of deterioration of soil quality (PLI > 1). The potential ecological risk (RI) in the soils of all the analysed urban zones of the city of Belgrade was <90, which indicates that these soils were of low ecological risk. The highest RI values were detected in the soils of the central zone (56.78) and the suburban zone (47.84), while they were significantly lower for the soils of the external (26.70) and rural (28.40) zones of the city.

Furthermore, it should be noted that a slightly higher value of this index occurring in the soils of the rural compared to the external zone of the city of Belgrade was the consequence of higher concentrations of Cd and Pb.

## 4. Discussion

*4.1. The Impact of Urbanization on the Distribution of Zn, Cu, Cd, Pb, Cr and Ni in the Soils of the Urban Zones of the City of Belgrade*

The results of the research focusing on the distribution of Zn, Cu, Cd, Pb, Cr and Ni in the topsoils and the subsoils of different urban zones of the city of Belgrade showed that the mean concentration values of these heavy metals increased in most locations from the rural towards the central zone (Figure 2), thus confirming the earlier findings of other authors that emphasise the decrease in the concentration of heavy metals along the urban–rural gradient [23,24,49,54,57].

The highest mean heavy metal concentration values in the topsoils and the subsoils were recorded in the central and the suburban zones of the city of Belgrade, and especially the higher concentrations of Zn, Cu, Pb and Ni. Those mean concentrations (of Zn, Cu, Pb and Ni) were higher than the MLV and the reference value. In the central zone of the city, the dense road network and the presence of the city's heating plants has a significant effect on the high mean concentrations. In the soils of the suburban zone, the highest mean concentration of those heavy metals originated mainly from industrial production sources, the number of vehicles on the roads, and the presence of the city's heating plants. These findings are in accordance with the findings by Biasioli et al. [23], Li et al. [33], and Lu et al. [60]. The long-term use of leaded fuel has contributed to the high concentrations of Pb in the topsoils and the subsoils of the central zone of the city of Belgrade [10,17,18,61]. Moreover, very high concentrations of Zn, Cu and Pb in the soils of the central zone indicate that the duration of pollution has a significant effect on their higher concentrations, which is also supported by the findings by Chen et al. [10], Foti et al. [24], Yang et al. [62], and Liu et al. [63]. Serbia is classified as a country with increased emissions of Pb into the atmosphere, which affects the contribution to the deposition of Pb from its own emissions, in comparison to other countries [64]. Namely, in 2004, emissions of Pb amounted to 40 t/year, in 2006 it was 60 t/year, while in 2010 it reached 147.7 t/year [65].

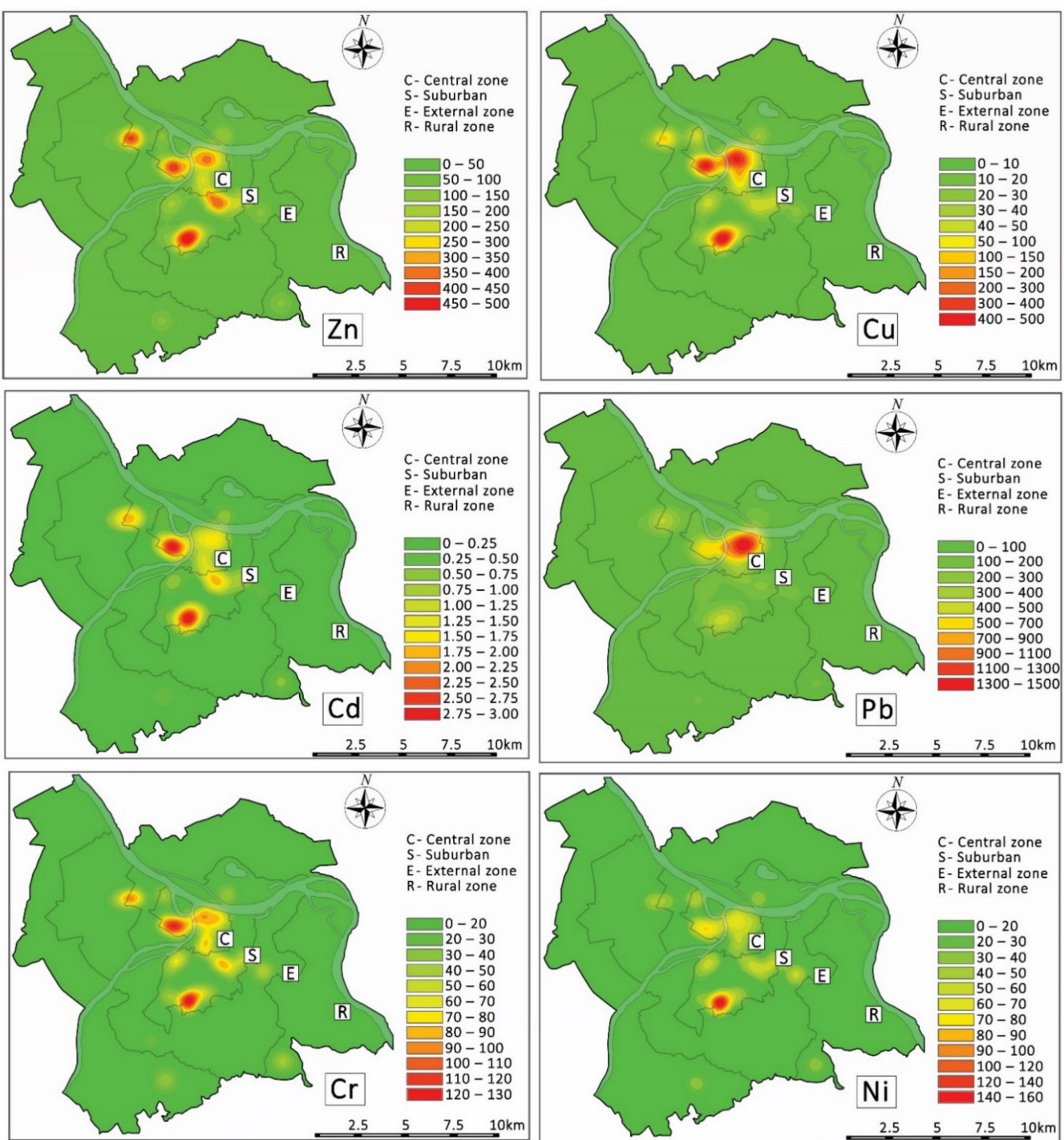

**Figure 2.** The mean concentrations of Zn, Cu, Cd, Pb, Cr and Ni (mg/kg) along urban–rural gradient of the city of Belgrade in both layers of the soils (topsoil 0–10 cm and subsoil 10–40 cm) of the urban zones (central, suburban, external, and rural) in the city of Belgrade.

The mean concentrations of Cu, Cd and Pb in the soils of the external and the rural zones of the city of Belgrade were below the reference value and the MLV, which is in accordance with the findings by Foti et al. [24] and Lee et al. [54]. The mean concentrations of Zn in all the analysed soils were above the reference value, which is a consequence of atmospheric deposition from the city's dense traffic network, the presence of industrial production and the burning of coal and other fossil fuels in individual households, which is supported by the findings of many authors [10,66,67] whose research have confirmed that

dry and wet atmospheric deposition represent major pathways of anthropogenic inputs of heavy metals into the soils.

The mean concentrations of Ni in the soils of all urban zones of the city of Belgrade were above the MLV, and above the reference value in the soils of the central and suburban zones. The mean concentration values of Cr were lower than the MLV in all the analysed soils of the urban zones of the city of Belgrade and higher than the reference value in the soils of the central and suburban zones. The findings concerning Ni and Cr were associated with the specific geological substrate of this area [58], which is supported by the findings of other research focusing on the soils in the city of Torino [23], Paris [24], and Beijing [63], etc.

The topsoils in urban areas are often enriched with metals particularly Zn, Cu, Cd, Pb, Cr and Ni, [1,62]. Several studies have confirmed that dry and wet atmospheric deposition represent major pathways of anthropogenic inputs of heavy metals into the topsoil [10,66,67]. For the highly urbanized and/or industrial urban areas, soil pollution and potential ecological risks should be considered a concern due to the high intensity of urbanization in developing countries such as Serbia.

*4.2. Possible Sources of Heavy Metals in the Soils of the City of Belgrade According to the Level of Urbanization*

Human activities have significantly contributed to the heavy metal accumulation in the soil for centuries, thus soil enrichment with heavy metals can reflect historical anthropogenic activities. Most anthropogenic pollutants are emitted into the atmosphere and then are deposited on the soil surface; however, the accumulation of metals may also be supported by natural processes [68]. The expression of the content of heavy metal differentiation within a soil profile can be seen in the pollution indices [68]. Pollution index values significantly depend on the geochemical background. A local geochemical background is characterized by the spatial-temporal changes conditioned by the content of heavy metals, which is usually a characteristic feature of a soil type or a region [69].

The results of the PCA test indicated that Cr and Ni in the soils of all urban zones of the city of Belgrade were distributed into a separate component: in the central and rural zones, they were distributed into PC1, and in the external and rural zones, they were distributed into PC2, while being highly correlated in these components (Figure 3). The possible common sources of Cr and Ni were also indicated by the results of the correlation matrix, which showed that they were significantly correlated ($p < 0.01$) in the soils of all four urban zones of the city. This was also confirmed by the fact that the concentrations of Cr and Ni are naturally increased as the result of a specific geological substrate for the city of Belgrade [58].

Namely, the research by Zhou and Wang [9] into the mean concentration values of heavy metals in three large urban agglomerations in China indicated that Cr and Ni belonged to the same component (PC1) in all three cities, which led to the conclusion that they had the same source—geological source. However, urban soils, as discussed earlier, are variable and under intense pressure from pollutants, thus the possible sources of Cr and Ni are not solely geological, but also anthropogenic, as indicated by the results of this research, which showed that these metals in the suburban and external zones dominated in the PC2 component. This was also supported by the results of the correlation matrix, which showed that in the suburban zone, Cr was significantly correlated with all the analysed heavy metals, while Ni had a low correlation only with Cd ($p < 0.05$) and was highly significantly correlated with Cr ($p < 0.001$). In the external zone, Cr was also correlated with Zn ($p < 0.05$), Cu ($p < 0.05$), and Ni ($p < 0.001$), while Ni was only correlated with Cr. In accordance with these data, are the results of the research on the presence of heavy metals in the soils and sediments along the Sava River basin [53], which indicated that Cr and Ni were distributed both in PC1 and PC2 components, i.e., that their presumed sources were influenced by both geological and anthropogenic factors. Additionally, the results of this research are supported by the study of Xia et al. [70], which showed that Cr and Ni were distributed in PC2, while Zn, Cu, Cd and Pb were distributed in PC1, as well as the

research by Madrid et al. [51], which showed that Cr and Ni were distributed in PC2, and Zn, Cu and Pb were distributed in PC1.

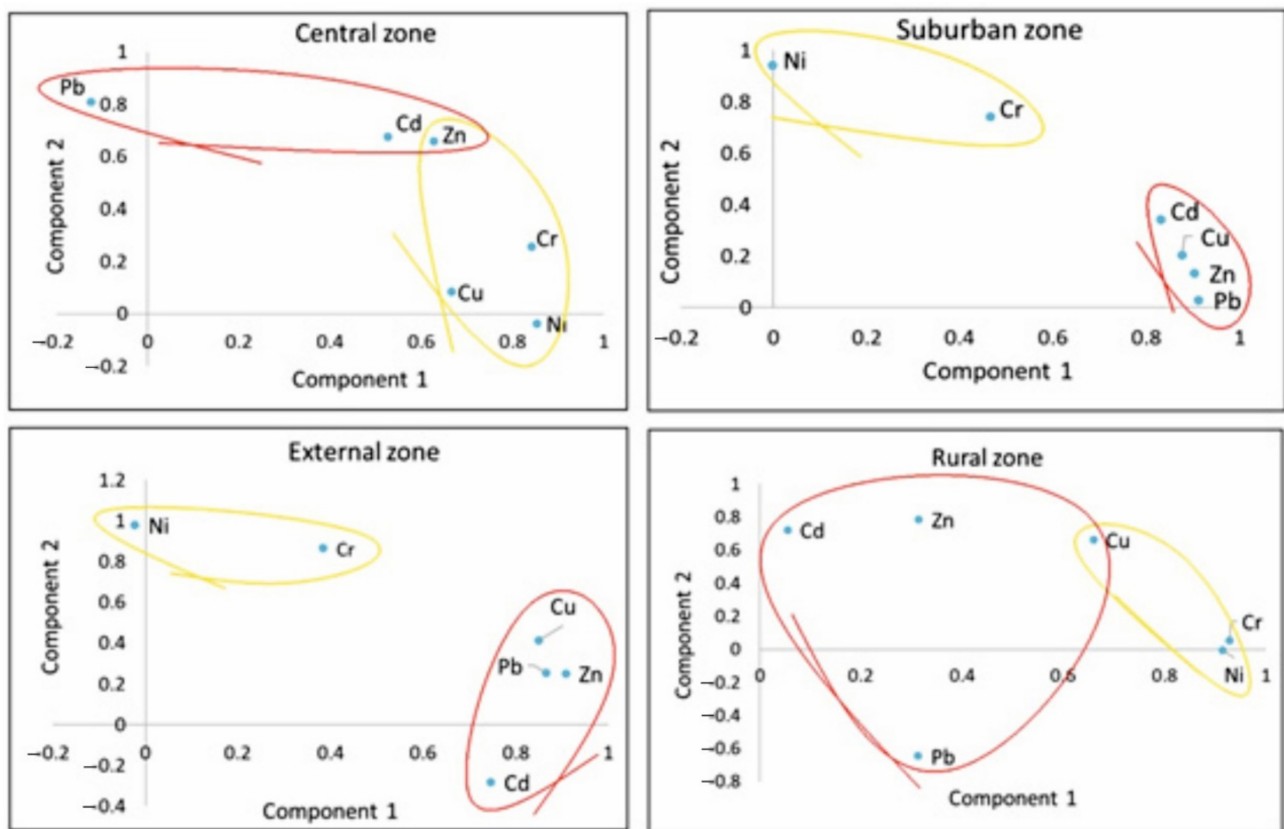

**Figure 3.** Possible sources of Zn, Cu, Cd, Pb, Cr and Ni in the soils of the urban zones of the city of Belgrade according to the results of the PCA test.

Together with Cr and Ni, Zn and Cu were distributed in the PC1 of the central zone, while Zn was distributed in both components. The correlation matrix indicated that Zn is highly correlated ($p < 0.001$) with Cu and Cr, and significantly correlated with Ni ($p < 0.001$), which may imply that their possible source in these soils is partly geological and partly belongs to anthropogenic sources of pollution. In the soils of the rural zone, Cu was also distributed in PC1 with Cr and Ni, and in PC2 with other metals. However, the correlation matrix of these soils did not show that Cu was significantly correlated with Cr and Ni in these soils. This result leads to the conclusion that the possible sources of Cu in these soils are largely due to anthropogenic sources of pollution, and much less of geological origin. The results of the PCA test showed that Cr and Ni were predominantly from geological sources. The same test shows that Zn, Cd and Pb were presumably a consequence of anthropogenic sources, as in all four urban zones, they were distributed within the same (PC2) component, while Cu can be from both geological and anthropogenic sources. All these results concur with the research by Zhou and Wang [9], Madrid et al. [51], Pavlović et al. [53], and Yang et al. [71].

*4.3. The Impact of Urbanization on the Level of Soil Pollution with Zn, Cu, Cd, Pb, Cr and Ni in Different Urban Zones of the City of Belgrade*

The Pollution index values derived from the mean heavy metals concentrations in the soils of the central zone of the city of Belgrade were the highest, and showed that Pb (2.46), Zn (1.77) and Cu (1.71) had the highest pollution potential, as well as the PI of Zn (1.96) in the suburban zone of the city. The PI values show that the mean heavy metals



concentration values decreased along the urban–rural gradient, which is in accordance with the previous research by Lu et al. [60].

The value of PI for Zn in the soils of the central and suburban zones of the city were at the upper limit of the pollution class II, and in the external zone it was at the lower limit of the same class, while in the soils of the rural zone of the city this value was at the upper limit of the pollution class I.

The research into the level of soil pollution in the city of Belgrade shows that the PI values for Cu in the central and suburban zones of the city were at the upper limit of the pollution class II. It was determined that in the external and rural zones of the city, there was no pollution with this heavy metal (pollution class I). Similar results were obtained concerning the urban soils of the city of Turin (Italy), which showed a high PI for Cu (within pollution class 4) [23]. The PI values for Cd showed that there was no soil pollution in the central and suburban zones of the city of Belgrade (pollution class I), although the values in these zones were at the upper limit of this class, while the PI values for Cd in the external and rural zones of the city of Belgrade were significantly lower. The PI values show that Cd and Pb pollution was slightly higher in the soils of the rural zone compared to the soils of the external zone of the city. The highest PI values for Ni were also recorded in the soils of the external zone compared to the soils of other urban zones of the city. Elevated values of these metals were associated with the features of the geological substrate. It was also determined that the PI values were the highest (at the lower limit of the pollution class III) for Pb in the soils of the central zone of the city. One of the biggest sources of Pb in urban soils is traffic, i.e., the use of leaded fuels. Although Pb fuel was phased out almost more than two decades ago, many studies still confirm that its concentration values remain high. These facts support the claim by Kabata-Pendias [59] that Pb is the least mobile element in soil. In addition to Pb, it was determined that the PI values for Cr and Ni were high in the soils of the central and suburban zones of the city (pollution class II), while they are lower in the external and rural zones (pollution class I). Similar results were obtained by Biasioli et al. [23], who determined the highest PI value for Pb (pollution class 5) and a lower PI value for Cr and Ni (both within pollution class III) in the urban zone of the city of Turin.

The determined $PI_{Nemerov}$, PLI, and RI indices show a similar distribution of soil pollution with the analysed heavy metals in relation to the degree of urbanization of the city of Belgrade. These data indicate the even distribution of elevated pollution indices throughout the territory of the city of Belgrade. The same results were obtained by Biasiola et al. [23] while researching the levels of soil pollution in the city of Turin. The $PI_{Nemerov}$ index showed that the soils of the central zone of the city of Belgrade belong to the pollution class III, that the soils of the suburban zone belong to the pollution class II, and that the soils of the external and rural zones belong to the pollution class I (Figure 4).

It is precisely this difference which indicates that the soils of the central zone of the city of Belgrade are polluted and that in the future this index should be expected to increase significantly. In support of this, is the fact that the PLI for the soils of all urban zones of the city of Belgrade fell into the deteriorating-in-quality category. The RI values show that all soils were of low environmental risk, given that the established values were less than 90. However, this should not diminish the concern, as the RI values of the soils in the central and suburban zones of the city were higher than the RI values of the soils in the external and rural zones of the city, which points to significant pollution levels. The results of this research concur with the research by Foti et al. [24] focusing on the soils of the city of Paris (France).

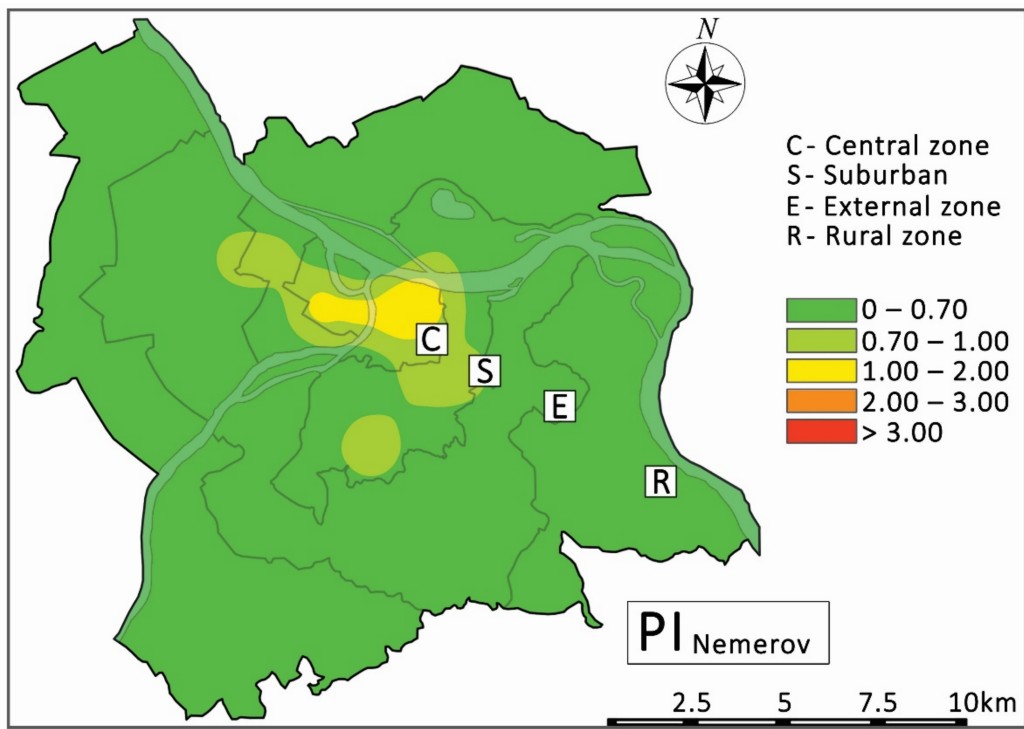

**Figure 4.** The map of soil PI$_{Nemerov}$ along the urban–rural gradient.

## 5. Conclusions

This study showed that the degree of urbanization significantly influences the concentration of Zn, Cu, Cd, Pb, Cr, and Ni in the soils of different urban zones (central, suburban, external, and rural) of the city of Belgrade. Furthermore, it was shown that the distribution of heavy metals in the soils of all urban zones of the city of Belgrade was heterogeneous and significantly higher in the central and suburban zones compared to the external and rural zones. The distribution of heavy metals soil pollution in most cases declined along the urban–rural gradient, i.e., it declined from the central zone towards the rural zone of the city.

Because of the presence of intensive traffic and the city's heating plants, in the central and suburban zones of the city, significantly higher mean concentration values of Zn, Cu and Pb were determined compared to their MLV and reference values. In addition, higher mean concentration values of Cd and Ni were determined, in comparison to their MLV, in the soils of the central zone of the city. The study also showed that the mean concentration values of Zn, Cu, Cd, Pb and Cr in the soils of urban zones of the city of Belgrade decreased with soil depth, except in the soils of the external zone for Cu and in the rural zone for Cu, Cd and Pb, while the mean concentration values of Ni increased with soil depth in all urban zones, because of a specific geological substrate characterized by elevated concentration of Ni and Cr.

The results showed that in the soils of different urban zones of the city of Belgrade, the possible sources of Cr and Ni were geological, and the possible sources of Zn, Cd and Pb were anthropogenic, while the mean concentration values of Cu were based on a variability controlled presumably both by anthropogenic and natural factors such as the geological substrate. Additionally, it was determined that the higher mean concentration values of Cr and Ni in the suburban and external zones of the city were probably from anthropogenic sources.

The indicators of heavy metals soil pollution showed a higher level of pollution in the central and suburban zones of the city of Belgrade, however, the general risk to the health of residents in the analysed urban zones was low. Based on the obtained mean heavy metals concentration values in the soils of different urban zones of the city of Belgrade, it was

concluded that they were not high, which indicates that the health risk to the environment of the city of Belgrade was relatively low. This fact was confirmed by the obtained values of $PI_{Nemerov}$ (1.27 to 1.44, 0.39), which indicates that they also belonged to the RI (1.18 and 26.70 and 56.78) indices. However, the results of the PLI values showed that the soils of all urban zones of the city of Belgrade were in the class II of the warning limit (0.7–1). This was especially noticeable in the soils of the central and suburban zones of the city, where the measured maximum concentration values of Zn, Cu and Pb were significantly higher even than the remediation values.

This study indicated that the soils of the central and suburban zones of the city of Belgrade were more intensely polluted with heavy metals (Zn, Cu, Cd, Pb, Cr and Ni), however, it was shown that the maximum concentrations of heavy metals in certain localities of the central and suburban zones were high, especially for Zn, Cu and Pb, which is primarily a consequence of intensive traffic.

These results indicate that the monitoring of the distribution and the level of pollution of urban soils with heavy metals (as one of the most dominant types of pollutants in the urban environment) is important to conduct by urban zones (the degree of urbanization), as it provides an applicable basis for the development of plans and strategies concerning urban soils use management for the purpose of urban environment protection and further sustainable urban development.

Understanding the complex relationship between the spatial distribution of heavy metals and the degree of urbanization (pertinent to specific urban zones of the city) is a prerequisite for reducing the negative impact of urbanization on the urban environment. Therefore, a better understanding of the relationship between urban spatial patterns and soil pollution can help determine the most efficient use of urban soils, which will ensure a healthier development of the city of Belgrade, but also other cities around the world.

The results of this study show that the impact of the level of heavy metals soil pollution the city of Belgrade on the quality of the urban environment should not be neglected, which points to the need to include monitoring of their concentrations in urban planning processes and urban management plans.

The results of this study show that there is a need to monitor the concentrations of Zn, Cu and Pb in the soils of the different urban zones of the city. It is also recommended to monitor the concentrations of Cr and Ni in the soils of the central and suburban zones of the city, as an increase in their concentrations was observed in these specific urban zones.

**Author Contributions:** Conceptualization M.T., P.P. and N.S.; methodology M.T., N.S., M.K. and P.P.; formal analysis and investigation, M.T.; writing—original draft M.T.; writing—review and editing N.S., M.K., P.P., D.Đ.-B. and J.P. All authors have read and agreed to the published version of the manuscript.

**Funding:** This work was supported by the Ministry of Education, Science, and Technological Development of the Republic of Serbia, grant no. 451-03-68/2022-14/200007 and 451-03-68/2022-14/200169.

**Institutional Review Board Statement:** Not applicable.

**Informed Consent Statement:** Not applicable.

**Data Availability Statement:** Data that support the findings of this study are available from the corresponding author upon reasonable request.

**Conflicts of Interest:** The authors declare no conflict of interest.

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
