# Peer review of "The Impact of the Degree of Urbanization on Spatial Distribution, Sources and Levels of Heavy Metals Pollution in Urban Soils—A Case Study of the City of Belgrade (Serbia)"

_sustainability, doi:10.3390/su142013126_

Round 1

Reviewer 1 Report

Line

16: Principal Component Analysis (PCA)

17: zinc (Zn), copper (Cu) cadmium (Cd), chromium (Cr) and nickle (Ni)

20-21: instead of repeating the letters twice perhaps just say “for the same elements”

48-54: Although you are writing about sources you use the word “sources” 7 times in the span of 4 sentences.

73-75: Check formatting and align to journal accepted formats.

128: surface of the city = land area, surface area, with proper units.

154: Instead of using bold fonts perhaps consider using subsections eg. 2.1.1 Suburban Zone, 2.1.2 External Zone, 2.1.3 Rural Zone, etc.

169-174: be consistent with the use of decimals or commas and significant digits. This would apply to the entire manuscript.

176: (0-10cm depth)

182-189: Rewrite this section, the English is not good.

184: composite samples were taken following the procedure outlined by [4]

186 by means of a stainless steel auger

193: Expand these element names as this is the first time they are mentioned.  Then use just the element symbols after. Zinc (Zn), Copper (Cu) Cadmium (Cd), Chromium (Cr) and Nickel (Ni) etc.

205: remove Cyrillic character

207: Move Figure 1 up to where it is first referenced.  Below line 181

224: The reference format changes at this point from [#] to Name [#], be consistent throughout.

227: Consider converting this table to a section in paragraph form.

245: Consider subsectons here instead of bold font. Eg. 3.1.1 Zinc Concentrations.

274: Change green font to black.

277 Reference Mao et al. 2014.  This should be [59] or Mao et al. [59] depending on which format you go with.

 286: Move table to line 245

310, 335: References

398-399: Remove Table refences on these lines. Insert (Table #4) following “The correlation matrix…” on line 400.

439: Move Table 5 down to line 461 to conform to what you did for Table 4.

516: reference sequence out of order. [64], then [63]… where is [62] there is one on line 604 ?

524:  remove the spacing between the words “analysed soils”

531: Move Figure 2 up to line 502 just after where it is referred to.

589: Move Figure 3 up to line 568 just after where it is referred to.

604: Yang reference out of sequence.

611: Refers to Figure 4, doesn’t appear to be a Figure 4 for PI.

614 Figure 5 refers to PINemerov

643: Refers to PINemerov perhaps this is where Figure 5 goes?

769: Citation is a different format, keep all references uniform in citation format.

783: Needs a space between Belgrade and 1999.

Author Response

Dear Sir or Madam,

Thank you for your suggestions. We have accepted all of them except:

The suggestion 154: Instead of using bold fonts perhaps consider using subsections eg. 2.1.1 Suburban Zone, 2.1.2 External Zone, 2.1.3 Rural Zone, etc.

Response 1: We believe that it would not be effective to divide the description of the zones into separate sections, given that they are described in only one or two sentences.

Sincerely

Reviewer 2 Report

The manuscript “The impact of the degree of urbanization on spatial distribution, sources and levels of heavy metals pollution in urban soils – A case study of the city of Belgrade (Serbia)” by Tešić et al. documents an important thing, soil metal contamination in Belgrade.  However, fundamental shortcomings in approach and interpretation prevent recommendation that this manuscript be published in this form.  This is a really cool data set and these critiques are offered as a way to strengthen the article, which will be very good in final form.

First: In general, the analyses are conducted from an overly aggregated perspective.  The manuscript is based on 126 sample locations, but all of the comparisons of metal content and urbanization are based on 4 aggregated zones.  This substantially diminishes the explanatory power of the dataset. Re-evaluation of the data based on local measures of urbanization (e.g., road density, impervious cover, etc.) compared to individual sites is strongly recommended. In particular, an inability to assess the location of the sample sites relative to a road greatly diminishes one’s ability to evaluate the role of historical gasoline additives to the soil loading.

Second:  There are fundamental misinterpretations of PCA results.  PCA loadings simply reorient the data so that it is organized along the gradient of most variability.  Similarities in positive loadings do not conclusively indicate similarities in source.  In particular, in this study sometimes the first two components only capture 50% of the variability, so the 3rd component could contain important detail about source.  In general, given the size of the data set PCA seems like an overly abstracted approach.  Bottom line, if the components are used to indicate source they must be carefully compared with potential sources.  That doesn’t appear to be included in the manuscript.

 Otherwise, less fundamental concerns are outlined below:

-The urban metal loading is considered “continuous” (ln 31, ln 35).  However, this contradicted by the information presented in the manuscript itself.  The increases in Pb flux in Belgrade indicate that the urban processes are NOT continuous.  These dynamics need to be incorporated into analysis and interpretation.  They likely will enhance and facilitate both.

-There are several cases where sweeping generalizations that are not necessarily accepted by the wider community are made based on single, very local studies (e.g., reference [1] ln 33, references [15] &[16] ln 54, reference [21] on line 66).  This is not acceptable.  Either make a stronger case or make the statements more nuanced.

-Line 51, 52 – this may not be always true, and may be not the main difference. This sentence should be written in different way.

- Line 114-120 Hypothesis – Can hypothesis 1, 4 and 5 be combined, they are saying the same thing.

-Line 128 – Surface or the area? What specific urban area?

-Lines 130-133 Too long sentence.  Break apart.

-Line 174  Consider adding information about predominant wind direction.

-Line 182- 189  Was the sampling random or … how did you choose the sampling places? Are they withing the residential gardening places? How close are they from the roads? Or industrial places?

-The accumulation of trace metals in organic carbon rich upper soil horizons is long recognized.  Calling it “current research” is misleading.  Also, the role of weathering needs to be incorporated into this discussion.

-Was a graphite furnace used when running the Hg?  If not, these data should probably not be used.

-ln 204-205: need to specify units, what does the "N" symbol stand for?

- Figure 1. Needs legend to explaing coloring differences.

-In general throughout the manuscript, 4 significant digits are reported.  This is too much precision.  Data should be presented with a more reasonable level of precision.

-Line 286 Table 3, need units.

-Throughout the results substantial discussion is included (e.g,. ln 253-261, 273-283).  Either reorganize into a results and discussion format or move the discussion content to the discussion.

-Line 274 Why is the reference author 54 in different color?

-A p-value tells one nothing about the strength of association.  The use of the p-value to justify a statement about strength of association (e.g., ln 409) is incorrect and should be revised throughout the manuscript.  Moreover, it’s not clear what the corr matrix was based on.  If it’s aggregated values the numbers of samples are very small and significance tests are suspect.

-Ln 458: One cannot infer a strong similarity in geologic source from component loadings without substantial additional justification.

-Line 580,  something wrong to this sentence, duplicate words.

-Line 584, omit the word ‘by’

-Line 624, Missing the close parenthesis.

-Figure 2:  How are these surfaces interpolated?  Need units, information about data (e.g., are they mean concentrations?)

-Data availability:  While requiring interested researchers to contact the authors may meet requirements for the journal it is not good practice for data curation in general.  I strongly suggest deposition of the data in a community data repository.  It makes the work so much more vital.

Author Response

Dear Sir or Madam,

Thank you for your suggestions. Below you can find our responses.

First: In general, the analyses are conducted from an overly aggregated perspective. The manuscript is based on 126 sample locations, but all of the comparisons of metal content and urbanization are based on 4 aggregated zones. This substantially diminishes the explanatory power of the dataset. Re-evaluation of the data based on local measures of urbanization (e.g., road density, impervious cover, etc.) compared to individual sites is strongly recommended. In particular, an inability to assess the location of the sample sites relative to a road greatly diminishes one’s ability to evaluate the role of historical gasoline additives to the soil loading.

Response 1: The focus here is on a different way of observing and interpreting research results. Certainly, the results of the research can be interpreted as reviewer 2 stated (based on local measures of urbanization e.g., road density, impervious cover, etc.), but then the goal of the research would be different. We investigated the impact of urbanization on soil pollution. We emphasized the impact of urbanization. As many cities today are, according to the degree of urbanization, divided into the so-called urban zones we interpreted our results in relation to them (which in a way gives a novel quality to the research).

The urban city zones are distinguished according to the degree of urban fabric built-up and the specificity of the urban morphology, i.e., according to the two central concepts that have a wide application in planning and management of urban areas. Also, the urban city zones are recognized as an indicator that represents an important tool for understanding the evolutionary process of urban areas, identifying the degree of urban expansion, predicting urban growth, etc., which can give our research results a wider application in practice.

Second: There are fundamental misinterpretations of PCA results. PCA loadings simply reorient the data so that it is organized along the gradient of most variability. Similarities in positive loadings do not conclusively indicate similarities in source. In particular, in this study sometimes the first two components only capture 50% of the variability, so the 3rd component could contain important detail about source. In general, given the size of the data set PCA seems like an overly abstracted approach. Bottom line, if the components are used to indicate source they must be carefully compared with potential sources. That doesn’t appear to be included in the manuscript.

Response 2: As the impact of urbanization on soil pollution was investigated in the paper, the results were interpreted in relation to the urban zones of the city, on the basis of which such organization of data was done. It certainly could have been done in a different way, but we thought that this way was more appropriate.

As in the research investigated the impact of urbanization on soil pollution, the results were interpreted in relation to the urban zones of the city, on the basis of which the data were organized in such a way. It certainly could have been done in a different way, but we thought that this way was more appropriate.

Also, we agree with the suggestion given by Reviewer 2 that in this way an abstracted approach to determining the heavy metals sources is presented. In order to act on this suggestion, in the paper we specifically indicated that determining the heavy metals sources is possible using the PCA test and the correlation matrix.

-The urban metal loading is considered “continuous” (ln 31, ln 35). However, this contradicted by the information presented in the manuscript itself. The increases in Pb flux in Belgrade indicate that the urban processes are NOT continuous. These dynamics need to be incorporated into analysis and interpretation. They likely will enhance and facilitate both.

Response: In the discussion section 4.1. data are given on the increase in the presence of Pb in the atmosphere in Serbia.

-There are several cases where sweeping generalizations that are not necessarily accepted by the wider community are made based on single, very local studies (e.g., reference [1] ln 33, references [15] &[16] ln 54, reference [21] on line 66). This is not acceptable. Either make a stronger case or make the statements more nuanced.

Response: The statements are nuanced.

-Line 51, 52 – this may not be always true, and may be not the main difference. This sentence should be written in different way.

Response: The suggestion is accepted.

- Line 114-120 Hypothesis – Can hypothesis 1, 4 and 5 be combined, they are saying the same thing.

Response: The suggestion is accepted

-Line 128 – Surface or the area? What specific urban area?

Response: The suggestion is accepted.

-Lines 130-133 Too long sentence. Break apart.

Response: The suggestion is accepted.

-Line 174 Consider adding information about predominant wind direction.

Response: The suggestion is accepted.

-Line 182- 189 Was the sampling random or … how did you choose the sampling places? Are they withing the residential gardening places? How close are they from the roads? Or industrial places?

Response: Sampling was conducted randomly.

-The accumulation of trace metals in organic carbon rich upper soil horizons is long recognized. Calling it “current research” is misleading. Also, the role of weathering needs to be incorporated into this discussion.

Response: We have rephrased ‘current research’. We agree that the role of weathering could be incorporated into this discussion, but this would call for significant changes in the research work. We certainly think that this is an interesting way of observing the results, which can be applied in a new research.

-Was a graphite furnace used when running the Hg? If not, these data should probably not be used.

Response: Тhis was a mistake, Hg wasn’t investigated in the paper.

-ln 204-205: need to specify units, what does the "N" symbol stand for?

Response: Units were mg/kg. N is actually a Cyrillic letter, thanks for pointing out that I didn't correct it.

- Figure 1. Needs legend to explaing coloring differences.

Response: The suggestion is accepted.

-In general throughout the manuscript, 4 significant digits are reported. This is too much precision. Data should be presented with a more reasonable level of precision.

Response: The suggestion is accepted.

-Line 286 Table 3, need units.

Response: The suggestion is accepted.

-Throughout the results substantial discussion is included (e.g,. ln 253-261, 273-283). Either reorganize into a results and discussion format or move the discussion content to the discussion.

Response: In these segments of the results, we compared our results with the results of other authors, because by repeating these data in the discussion section, we believe that we would burden the section.

-Line 274 Why is the reference author 54 in different color?

Response: The suggestion is accepted.

-A p-value tells one nothing about the strength of association. The use of the p-value to justify a statement about strength of association (e.g., ln 409) is incorrect and should be revised throughout the manuscript. Moreover, it’s not clear what the corr matrix was based on. If it’s aggregated values the numbers of samples are very small and significance tests are suspect.

Response: We wrote this segment in a different way.

-Ln 458: One cannot infer a strong similarity in geologic source from component loadings without substantial additional justification.

Response: We agree. Here we are focusing on a possible source, which has been changed in the entire text.

-Line 580, something wrong to this sentence, duplicate words.

Response: The suggestion is accepted.

-Line 584, omit the word ‘by’

Response: The suggestion is accepted.

-Line 624, Missing the close parenthesis.

Response: The suggestion is accepted.

-Figure 2: How are these surfaces interpolated? Need units, information about data (e.g., are they mean concentrations?)

Response: The suggestion is accepted.

-Data availability: While requiring interested researchers to contact the authors may meet requirements for the journal it is not good practice for data curation in general. I strongly suggest deposition of the data in a community data repository. It makes the work so much more vital.

Response: We agree with the reviewer's opinion. In some future period, we could deposit our results in a community data repository.

Sincerely

Reviewer 3 Report

The scientific approach has elements of originality in solving the research theme.

Methodology is appropriate.

The results are presented clearly and in a logical sequence.

Figures and tables accurately describe data in the article.

Conclusions are formulated concisely and scientifically argued.

Author Response

Dear Sir or Madam,

Thank you for your positive review. 

Sincerely

Round 2

Reviewer 2 Report

While I find the underlying data to have exciting potential, I find the authors response to be insufficient.

For example, my first major critique is the analysis uses data from an over aggregated perspective.  I make some suggestions on other things to examine.  The authors response is to discuss why my suggestions are wrong, but they never address the over aggregation which is the most important part of the critique.  They can make more precise characterization of their urbanization measures (e.g., urban fabric).  They, however, do not specify why they don't.

I don't think the authors took the review seriously.

Author Response

Dear Sir or Madam,

We are sorry that you think we did not take your review seriously. It is probably our fault that we did not provide more detailed explanations, which we will try to correct below.

Why do we use urban zones of the city?

Defining and differentiating an urban zone as a term in relation to the degree of urban development and urban morphology are two central concepts that have a wide application in planning and management of urban areas. The transition between a high-rise and densely built, strictly urban center and a suburban one often becomes unclear and dispersed. Urban and suburban environments are characterized by a wide diversity of land use, which is expressed in a complex, diverse and highly fragmented landscape morphology. Suburban zones consist of a mosaic of diverse land cover, buildings and traffic infrastructure. Intensive expansion of the urbanized area has resulted in different configurational types of urban development, and significant attention has been devoted to research on the dynamic development of urban morphology from the perspective of landscape ecology. As an important indicator of landscape ecology, spatial patterns of urban growth refer to the locations of newly created fragments in developed landscapes. This indicator (urban zone of the city) is a powerful tool for understanding the evolutionary process of urban areas, identifying the degree of urban expansion and predicting urban growth. This is exactly why many cities today are, according to the degree of urbanization, divided into the so-called specific urban zones, i.e., parts of the city (segments of the urban fabric) that, according to the urban development intensity, are characterized by a similar urban morphology. Urban zones can be parts of megalopolises, urban areas, cities, suburbs or residential areas which were created by changing the appearance of the land/landscape through an increase in the number of people and/or built structures (by reducing green areas).

How did we identify the urban zones of the city?

The urban zones of the city of Belgrade were taken from the Master Plan of the city of Belgrade, which is an official document of the Republic of Serbia (RS) and was adopted according to the Law on Planning and Construction ("Official Gazette of RS", No. 72/09, 81/09, 64/10, 24/ 11, 121/12, 132/14 and 145/14), and the Statute of the City of Belgrade ("Official Gazette of the City of Belgrade", No. 39/08, 6/10, 23/13 and "Official Gazette of RS", No. 7/16 – CC Decision).

According to the Master Plan of the city of Belgrade, the division into urban zones (units) was made on the basis of the analysis and assessment of the entire territory of the city plan, which included the existing land use, typological, morphological, ambient and other relevant characteristics of space and content elements, as well as planning and strategic guidelines concerning urban development of the City of Belgrade. Consequently, on the basis of this data a basic zoning proposal was defined.

The boundaries of the urban zones are defined based on: - zonal division from the Master Plan of the city of Belgrade 2021; - statistical territorial units and circles; - characteristic urban units and sub-units; - re-examination of detailed urban development plans; - the prevailing characteristics of the purpose, typology of construction and methods of spatial organization and planning; - official statistical data (concentration and density of population, the employed, residential and business space, etc.); - proposal for the boundaries of the building/construction areas; and - planning and strategic solutions concerning the future development of the city.

According to the Master Plan of the city of Belgrade 2021 (MPB 2007), four urban spatial units are defined, which make up the main spatial-functional, urban-architectural and landscape-ecological zones of the City of Belgrade, namely: the central zone, the suburban zone, the external zone and the rural zone.

Sincerely

Round 3

Reviewer 2 Report

I'm sorry.  My previous comments were meant for the editor.  Editors make decisions on content.  I feel like less aggregation would make your study stronger and you remain unconvinced.  The decision on whether you need to make changes is one for the editor.

I apologize that you were caught in the middle on this.  This is not a quality review process for either of us.  I will step back at this point.